# Potential Use of Plant Growth-Promoting Bacteria to Enhance Growth and Soil Fertility in Marginal Areas: Focus on the Apulia Region, Italy

Angela Racioppo *, Annalisa d'Amelio, Alessandro De Santis, Antonio Bevilacqua, Maria Rosaria Corbo and Milena Sinigaglia *

Department of Agriculture, Food, Natural Resources and Engineering (DAFNE), University of Foggia, Via Napoli 25, 71122 Foggia, Italy; annalisa.damelio@unifg.it (A.d.); alessandro.desantis@unifg.it (A.D.S.); antonio.bevilacqua@unifg.it (A.B.); mariarosaria.corbo@unifg.it (M.R.C.)

* Correspondence: angela.racioppo@unifg.it (A.R.); milena.sinigaglia@unifg.it (M.S.)

**Abstract:** Soil degradation is a global problem and refers to the reduction or loss of the biological and economic productive capacity of the soil resource. In Europe, the countries most affected by soil degradation are undoubtedly those of the Mediterranean basin. Among these, Italy shows clear signs of degradation, with different characteristics, especially in the southern regions, where climatic and meteorological conditions strongly contribute to it. Apulia, the Tavoliere plain in particular, is a fragile and very sensitive ecosystem due to its intrinsic characteristics and the level of anthropic exploitation. Agricultural production pays the highest price, as increasing desertification due to climate change and the loss of agricultural land severely limit the extent of land available to produce food for an ever-growing population. Plant growth-promoting bacteria (PGPB) could be a low-cost and long-term solution to restore soil fertility, as they provide a wide range of benefits in agriculture, including increasing crop productivity, improving soil nutrient levels and inhibiting the growth of pathogens. This review shows how PGPB can be used to improve the quality of soils, their impact on agriculture, their tolerance to abiotic stresses (drought, salinity, heavy metals and organic pollutants) and their feasibility. The use of PGPB could be promoted as a green technology to be applied in marginal areas of Apulia to increase soil fertility, reduce pollution and mitigate the impacts of abiotic stresses and climate change. This is supported by a series of studies showing that the growth of plants inoculated with PGPB is superior to that of non-inoculated plants.

**Keywords:** marginal lands; plant growth-promoting bacteria; salinity stress; drought stress; pollution; sustainable production



## 1. Introduction

In agriculture, terms such as degraded, underutilised, fallow, desolate and forbidden are often used to define marginal land. However, the definition of marginal land depends on the intended use of the land and the context in which it is applied [1], though generally, marginal agricultural land refers to soils of low quality, characterised by low productivity and inadequate agricultural yields. There are many reasons why soils are defined as marginal, including the lack of water supply, low soil chemical and/or microbiological quality, pollution from previous industrial activities, topographical obstacles such as an extreme slope or inaccessibility. In addition, there may be contamination by heavy metals (HMs) and organic pollutants, strong acidification or alkalinisation, high salinity, etc. [2,3]. Biophysical and socio-economic aspects are the two central dimensions of agricultural marginality; indeed, although the lack of access to markets and services is an important element in determining the overall condition of marginality, the biophysical conditions are one of the main factors, especially from the perspective of crop production. Marginal lands are increasing due to the decline in natural and semi-natural ecosystems, the severe climate

change and human activities. While climate crisis, pollution and harmful deforestation practices are being increasingly discussed, few investigators address and describe a crisis of equal magnitude, i.e., global land degradation. As a result, there is a risk that food production will not be sufficient to feed the world's growing population. According to the latest United Nations forecasts, the global population could reach 8.5 billion by 2030, 9.7 billion by 2050 and 10.4 billion by 2100 [4]. We are therefore losing land at a time when we should be increasing agricultural production.

Remedial action is needed to address this major challenge and prevent the degradation and abandonment of land on which 98% of the world's food is produced. The loss of soil fertility is one of the main environmental threats; for years, the low yields on marginal land have been solved by the overuse of fertilisers, which has had a negative impact on both soil and health. Finding sustainable solutions that would enable crops to cope with abiotic and biotic stresses and promote their development on low-fertility soils is therefore the goal of modern agriculture. In this scenario, one of the most promising and particularly sustainable research areas is that of biostimulants. According to the current regulatory framework in Italy, biostimulants for agricultural use fall under the product category of "fertilisers" and are regulated by Legislative Decree No. 75 of 29 April 2010 (L.D. 75/2010). Biostimulants, according to Legislative Decree 75/2010, belong to the category of "products with specific action", defined as "products that add to another fertiliser, to the soil or to the plant, substances that promote or regulate the uptake of nutrients or correct certain abnormalities of a physiological nature, the types and characteristics of which are listed in Annex 6" [5].

In Europe, like in Italy, the decision was taken to include biostimulants in the fertiliser category, along with fertilisers, correctives, soil conditioners and growing media.

Regulation (EU) No 2019/1009 [6], which came into force in July 2022, defines plant biostimulants as "fertilising products (substances and/or microorganisms) whose function is to stimulate the nutritional processes of plants, irrespective of the nutrient content of the product, with the sole purpose of improving nutrient use efficiency, tolerance to abiotic stresses, quality characteristics or increasing the availability of nutrients locked up in the soil or rhizosphere" [7]. This definition is linked to the clarification that biostimulants are fertilisers and not plant protection products. The text also defines the component materials that can be used to produce biostimulants, which are "plants, plant parts or plant extracts" or "micro-organisms". These categories of materials are called "Component Material Categories" (CMCs). Fourteen CMCs are defined in the Regulation. Some changes were made to include new raw materials. Regarding microorganisms (CMC 7), only four are listed in the Regulation: *Azotobacter* spp., mycorrhizal fungi, *Rhizobium* spp. and *Azospirillum* spp. [7]. The biostimulatory effect of plant growth-promoting bacteria (PGPB) is well documented and has been studied for individual bacteria in microbial consortia and in complexes with organic matrices. PGPB are able to fix atmospheric nitrogen, solubilise useful plant elements such as phosphorus and iron and produce phytohormones such as auxins, gibberellins (GAs), cytokinins (CTK) and ethylene (ETH). In addition, these bacteria improve plant tolerance to various stresses such as drought, high salinity, metal toxicity and the effects of pesticides [8]. Many bacteria that have beneficial effects on plants belong to the genera *Azotobacter*, *Acinetobacter*, *Bacillus*, *Pseudomonas*, *Arthrobacter*, *Azospirillium*, *Rhizobium*, *Serratia*, etc.

Italy is a territory highly vulnerable to various degradation factors due to erosion and disaggregation, salinisation, contamination (local and diffuse), decline in organic matter, loss of biodiversity and land consumption. The situation has been aggravated in recent years by the quantitative and qualitative increase in droughts. The Apulia region is home to areas facing significant challenges, including high hydrogeological risks, declining soil fertility, rapid urbanization, desertification and a heightened threat of erosion. These factors have contributed to the abandonment of agricultural lands in the region. The abandonment phenomenon mainly affects the less fertile areas and those located in mountainous terrain and/or characterised by poor infrastructure, in particular the hilly areas of the region, from

the Dauno Apennines to the Murgia plateau, from Salento to the Gargano peninsula. This part of Italy suffers from high temperatures (in the autumn/winter period) and a decrease in rainfall. A further threat to Apulia crop production may come from rising temperatures and, consequently, drier conditions (in summer) caused by the increasingly evident effects of climate change [9].

Given the current need to combat soil degradation and promote high-quality agricultural production by reducing the use of fertilisers, pesticides and water, the use of PGPB is a good sustainable strategy to promote plant development on degraded and fallow land [10]. The main objective of this review was to provide a general overview of PGPB and to discuss their efficacy and role as biological tools to promote the development and resistance of crop plants in soils affected by physical diseases and chemical and biological degradation processes. Thus, we will highlight the potential of PGPB to increase production yields and promote plant development, with the possibility to reintroduce crops that were lost due to soil degradation in marginal areas. This aspect is very interesting for the marginal areas of Apulia; to our knowledge, there are still few studies in the literature on the use of PGPB in these areas. At the same time, however, this could be a model that could be applied in other areas of the world that are affected by the same problems.

## 2. PGPB: Their Role as Growth Promoters and Biofertilisers

It is well known that PGPB play an important role in the agricultural ecosystem by improving soil fertility, promoting plant growth and inducing tolerance to biotic (phytopathogens and parasites) and abiotic (drought, salinity and HM) stresses through a variety of mechanisms. These beneficial bacteria can stimulate plant morphological (plant growth and yield), physiological and biochemical (photosynthesis, pigmentation, osmotic adjustment and antioxidant mechanisms) and metabolic processes by establishing positive and mutualistic plant–microbe interactions in the soil [11,12]. PGPB can colonise both the rhizosphere (the soil region surrounding the roots) and the root surface or associated tissues [13]. They can be free-living bacteria, symbionts (forming specific symbiotic relationships with plant roots), endophytes (colonising part or all of plant internal tissues) and cyanobacteria (formerly known as blue-green algae) [14]. The mechanisms by which PGPB affect plant performance are of two types: direct and indirect (Figure 1).

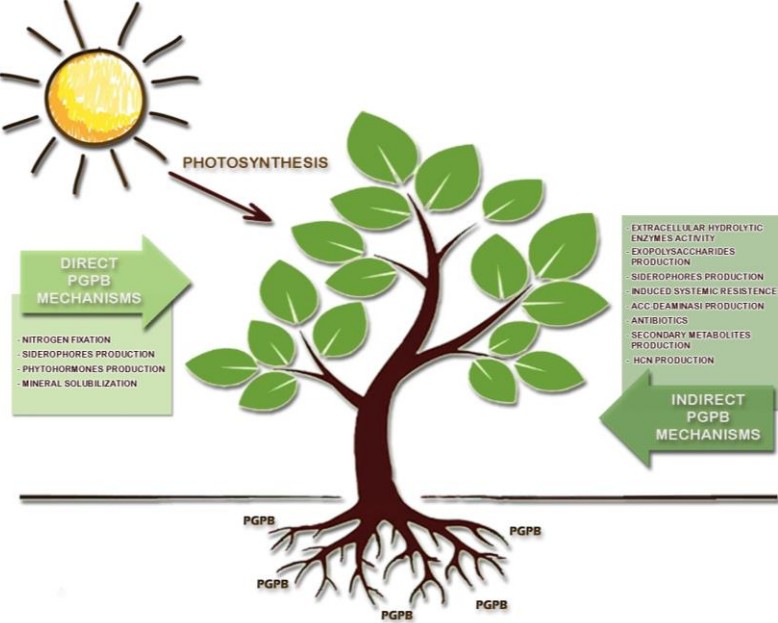

**Figure 1.** Direct and indirect PGPB mechanisms to support plant growth.

Nitrogen fixation for nitrogen supply through symbiotic and non-symbiotic mechanisms is an essential function of PGPB to increase nutrient availability. Furthermore, as shown by Romero-Munar et al. [15], PGPB can facilitate the uptake of other nutrients, such as potassium (K), by regulating root $K^+$ transporters. PGPB are also known to deliver and control several plant hormones GAs, CTK, abscisic acid (ABA), ETH and indole-3-acetic acid (IAA) to stimulate plant growth and development, including cell elongation, cell division, root development, root hair formation, shoot initiation and tissue differentiation [16,17].

The indirect mechanisms include the inhibition of various pathogens or the prevention of plant disease effects, through the production of metabolites including antibiotics, siderophores, volatile organic compounds (VOCs), hydrolytic enzymes, hydrogen cyanide (HCN) and 1-aminocyclopropane-1-carboxylic acid deaminase (ACC-deaminase). These metabolites can reduce or prevent pathogenic diseases and protect against environmental stresses [18,19]. One strategy used by microorganisms to compete with other microorganisms is the synthesis of low-molecular-weight antibiotics. Several antibiotics can be produced by different microorganisms; for example, bacteria belonging to the genus *Bacillus* are known to produce various antibiotics, such as iturins, mycosubtilin, bacillomy-cin D, surfactin, fengicin and zwittermycin A [20].

Another indirect mechanism involves the ability of PGPB to produce exopolysaccharides (EPSs) and form biofilms [21]. EPSs are important for the development of bacterial biofilms, as they are responsible for the adhesion of bacteria to soil particles and root surfaces [21,22]. Biofilms can help the host plant to grow, reduce microbial competition and provide protection against pathogens and abiotic stresses [23].

In the last few years, a number of interesting articles were published on the biostimulant and biocontrol effects of PGPB on plants. Table 1 shows the most important results of the effects of PGPB on major Apulian crops.

**Table 1.** PGPB and their biostimulant effects on plants. * (V: in vitro condition; F: field condition; Ghouse: greenhouse condition; planta: planta condition).

| PGPB | Plant Name | PGPB Mechanisms | Application Method * | Role of PGPB | Reference |
|---|---|---|---|---|---|
| *Brevibacterium casei* *Pseudomonas oryzihabitans* *Bacillus aryabhattai* | *Salicornia europaea* | ✓ Nitrogen fixation ✓ IAA, EPS$_s$ production ✓ ACC deaminase | Seed and soil inoculation (V, F) | ✓ Increased plant biomass ✓ Improved development of the root system and aerial parts of the crop | [24] |
| *Bacillus* spp. | *Solanum lycopersicum* | ✓ P solubilisation ✓ IAA production | Seed inoculation (Ghouse) | ✓ Improvement of seed germination ✓ Increased stem and root length | [25] |
| *Azospirillum brasilense* | *Zea mays* L. | ✓ Nitrogen fixation ✓ IAA, phytohormones production | Seed and/or foliar spray inoculation (Ghouse, F) | ✓ Improved root system ✓ Increased yield | [26] |
| *Bacillus* spp. *Pseudomonas* spp. *Streptomyces* spp. | *Z. mays* L. | ✓ P solubilisation ✓ Nitrogen fixation ✓ IAA, siderophore production | Seed inoculation (F) | ✓ Increased shoot root length and growth of the plant | [27] |
| *Rhizobium tropici* *Azospirillum brasilense* | *Phaseolus vulgaris* L. | ✓ Nitrogen fixation ✓ IAA, GA$_3$, CTK, ETH production | Seed and/or foliar spray inoculation (F) | ✓ Increased plant growth and yield and increased amount of N accumulated in root nodules | [28] |
| *Bacillus subtilis* *Bacillus amyloliquefaciens* *Bacillus megaterium* *Bacillus licheniformis* | *S. lycopersicum* L. | ✓ Auxin production ✓ Nitrogen fixation | Soil inoculation (F) | ✓ Increased plant growth and yield ✓ Improved plant physiology and quality characteristics | [29] |
| *Enterobacter* spp. *Pseudomonas* spp. | *S. lycopersicum* L. | ✓ AA, siderophores production ✓ P-solubilisation | Seed and soil inoculation (V) | ✓ Improved plant physiology ✓ Increased root and shoot dry biomass | [30] |
| *Azospirillum brasilense* | *Z. mays* L. | ✓ IAA, siderophores production ✓ ACC deaminase | Seed inoculation (F) | ✓ Increased plant growth and yield ✓ Increased plant N content | [31] |

**Table 1.** *Cont.*

| PGPB | Plant Name | PGPB Mechanisms | Application Method * | Role of PGPB | Reference |
|---|---|---|---|---|---|
| *Azospirillum* spp. | *Solanum tuberosum* | ✓ Nitrogen fixation<br>✓ IAA production<br>✓ P solubilisation | Tuber Inoculation (Ghouse, F) | ✓ Increase shoot and root length<br>✓ Increase plant biomass<br>✓ Increase N content | [32] |
| *Bacillus circulans (GN03)* | Cotton (*Gossypium hirsutum*) | ✓ Phytohormones production | Soil inoculation (Ghouse) | Accumulation of<br>✓ Growth-related hormones (indoleacetic acid, gibberellic acid, and brassinosteroids);<br>✓ Disease resistance-related hormones (salicylic acid and jasmonic acid)<br>Regulation of gene expression<br>✓ phytohormone synthesis-related (EDS1, AOC1, BES1 and GA20ox) genes;<br>✓ auxin transporter (Aux1);<br>✓ disease resistance (NPR1 and PR1) | [33] |
| *Azotobacter vinelandii* (encapsulated in alginate-Na beads) | *S. lycopersicum* L. | ✓ Nitrogen fixation<br>✓ IAA, GA$_3$, auxin, vitamins, amino acids production<br>✓ HCN production | Soil inoculation (Ghouse) | ✓ Increase shoot and root length<br>✓ Increase plant biomass<br>✓ Increase N content | [34] |
| *Rhizobium* (SP20, N8, N9, G56, G58, B02) | Cotton (*Gossypium hirsutum*) | ✓ P solubilisation | Soil inoculation (Ghouse) | ✓ Improvement seed germination<br>✓ Increased plant biomass<br>✓ Increase shoot and root length | [35] |
| *B. subtilis MBI600* | *Lycopersicum esculentum* | ✓ ETH, salicylic acid (SA) production<br>✓ Induction of systemic resistance (ISR) | Soil inoculation (V, planta) | ✓ Increase shoot and root length<br>✓ Activation of two auxin-related genes (SiPin6 and SiLax4)<br>✓ Biocontrol efficacy against pathogens | [36] |

### 3. Role of PGPB in Abiotic Stress Reduction

Agricultural productivity in marginal areas is affected by several environmental stresses, which can be divided into abiotic and biotic stresses. Salinity, drought, flooding, HM contamination, temperature extremes and pH are the major abiotic stresses. PGPB are known to alleviate the negative effects of stress on plants by influencing their stress response processes. Based on scientific evidence, it can be stated that the use of PGPB formulations is beneficial for plant development and is a way to transform damaged and fallow soils into healthy ones [10]. After focusing on the general mechanisms of action of PGPB, this section provides an overview of the specific effects of some of these mechanisms on biotic and abiotic stresses (drought, salinity and soil contamination) that affect agricultural productivity in marginal areas (Figure 2).

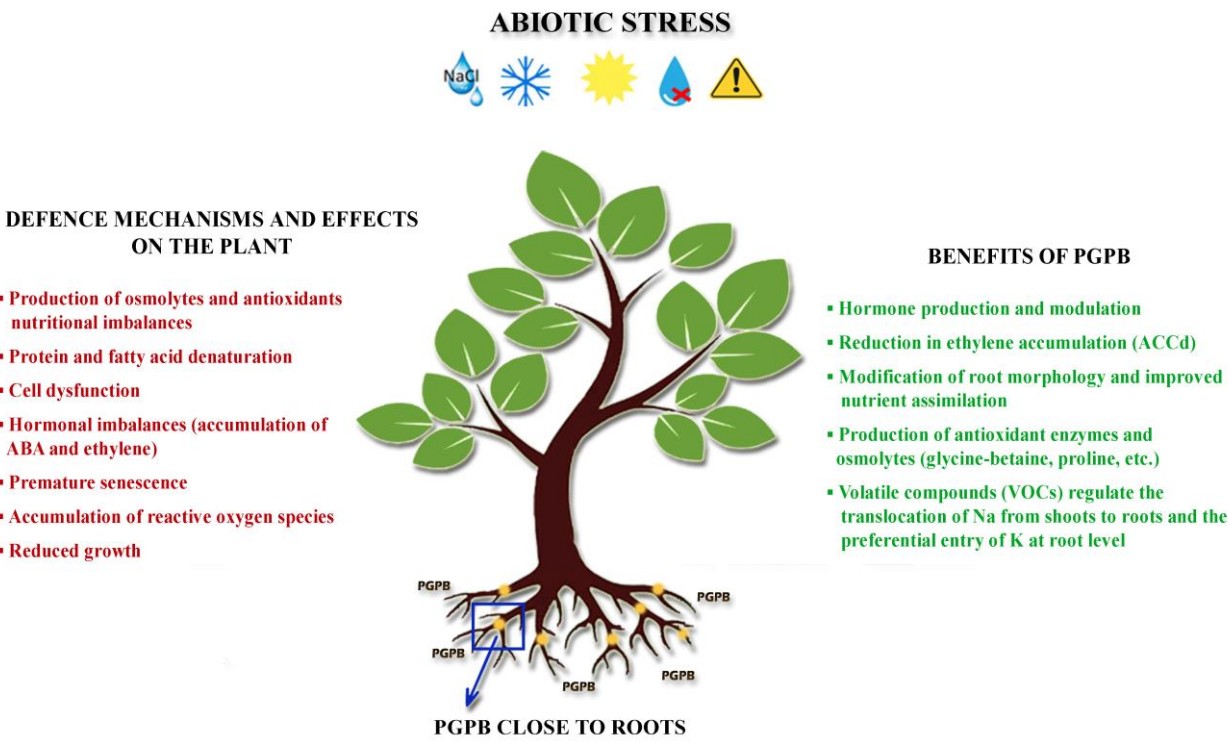

**Figure 2.** Plant defence mechanisms and beneficial effects of PGPB under stressful conditions.

### 3.1. PGPB—Plant Growth Promoters in High-Salinity Soils

Salinisation is one of the problems faced by marginal soils for a variety of reasons. Soils can be affected by natural or secondary salinity. In the former case, salt enrichment is often inherited from the material from which the soil is derived (igneous rocks of the lithosphere) and subsequently promoted by climatic conditions and hydrological events that favoured the deposition of large amounts of salts in sedimentary rocks, surface and subsurface waters, seas and oceans [37]. Natural saline soils are also found along marine coastlines, where infiltration of surface water tables and/or marine aerosols enrich soils with salt (NaCl) over a range of several hundred metres to a few kilometres. Secondary salinisation, on the other hand, results from the careless anthropogenic management of soils that are already vulnerable to this threat, with the use of inappropriate water and irrigation methods and inappropriate fertilisation, the advancement of the salt wedge due to over-exploitation (and misuse) of groundwater, and inadequate soil drainage conditions [38]. In Italy, salinisation affects many lowland areas, particularly coastal areas. Among the southern regions, Apulia, Sicily and Sardinia (to a lesser extent, Basilicata, Calabria and Campania) are the areas

most affected by this phenomenon. In Apulia, the most exposed areas are the coastal ones, in particular the Gargano, the Murge Baresi, the Salento, the Ionian–Taranto arc and the Adriatic coast. These areas are characterised by the presence of high concentrations of salts caused by the overexploitation of coastal aquifers for agricultural, industrial and civil purposes. The use of water for irrigation has a negative impact on soil fertility and production, in terms of both yield and product quality.

Tolerance to moderate salinity (4–8 dS m$^{-1}$) is a characteristic of Mediterranean plants, with some species showing sensitivity to salt stress but good adaptation to drought [39]. For example, *Ficus carica* L. has moderate salt tolerance, and growth under saline conditions does not result in a major reduction in biomass but is characterised by a reduction in relative water and chlorophyll content [40]. The growth of *Beta vulgaris* L. and *Sacharum officinarum* L. was increased under conditions of moderate salinity, but their photosynthetic activity and nutrient uptake were limited [41]. Furthermore, reduced flower and seed production under salinity conditions was observed in some Mediterranean crops such as chickpea and grapevine.

Salinity reduces the ability of plants to absorb water from the soil (osmotic stress) and leads to increased concentrations of ions such as Na$^+$ and Cl$^-$ in cells, which can exceed the toxic thresholds (ionic stress) [42]. Osmotic and ionic stress reduces cell expansion and causes nutrient imbalances and oxidative stress that affects plant growth, development and survival [43]. Due to the presence of excess soluble salts, plants struggle to absorb water from the soil because the circulating solution is so concentrated that it creates a high osmotic potential, as a result of which plant roots, instead of absorbing water, release it, causing dehydration. This phenomenon inhibits key plant metabolic processes, such as photosynthesis, protein synthesis and lipid metabolism, and adversely affects productivity. Several strategies have been implemented to combat salt stress, including the use of salt resistance genes in conventional crops, but these are only effective under laboratory conditions. Another approach is the pre-treatment of biological materials with specific and selective chemicals such as ascorbic acid, nitric oxide, phosphoric acid and glycine betaine [44] or with physical effectors such as UV-B irradiation [45]. However, although effective, these treatments are not recommended for sustainable agriculture. A possible sustainable solution could be the use of soil bacterial and fungal communities that colonise plant roots and stimulate their growth (PGPB). Although studies in the Mediterranean, and particularly in the Apulian context, are very scarce, a recent study demonstrated the beneficial effect of PGPB on durum wheat under drought and water stress conditions, improving photosynthetic efficiency, grain yield and plant height [46]. Under saline conditions, bacterial inoculation consistently improved nutrient uptake and increased plant biomass compared to non-inoculated plants. The ability of PGPB to improve the growth and yield of many crops, some of them typical of the Apulian context, grown under saline conditions outside the Apulian region, was reported in several studies. Bacterial inoculation was also shown to improve photosynthetic parameters in sugar beet [47,48], and similar results were obtained in tomato, rice and wheat [49,50]. PGPB can alleviate salinity stress by modifying stress-induced physiological changes in plants through various mechanisms, such as the regulation of the synthesis of various phytohormones, including IAA, ACC-deaminase, EPSs and volatile organic compounds, atmospheric nitrogen fixation, nitrogen solubilisation and the solubilisation of mineral phosphate [16,51]. Therefore, harnessing the potential of PGPB could improve crop performance in saline soils [52].

The main strategy used by plants to tolerate salt stress is the translocation of sodium into vacuoles, thereby reducing the amount of sodium in the cytoplasm. A recent study showed that PGPB increase the expression of genes encoding the plasma membrane protein salt overly sensitive exchanger 1 (SOS1) and other proteins related to the SOS pathway [53]. Similarly, in wheat plants subjected to salinity stress and treatment with *Dietzia natronolimnaea*, a significant increase in the expression of SOS1 (localised on the plasma membrane) and SOS4 was observed compared to non-inoculated plants [54]. The process of adaptation to stress involves most metabolic processes in plants, but it is generally accepted that

plant hormones such as ABA, SA acid and ETH play an important role in activating the signalling cascade associated with several genes involved in enhancing salt tolerance [55]. Research conducted on tomato under salt stress condition showed that the use of two *Bacillus* species (*B. aryabhattai* H19-1 and *B. mesonae* H20-5) increased carotenoid content, proline and ABA levels and antioxidant enzyme activities [56]. ACC-deaminase production by PGPB is the most studied mechanism by which plants alleviate salinity stress in soils by reducing the levels of ETH precursors, which tend to increase in plants under stress [57]. ACC-deaminase-producing *Pseudomonas* spp. promoted the growth of *Citrus macrophylla* under salinity stress conditions, contributed to the accumulation of IAA in leaves and inhibited the accumulation of chloride and proline in roots [58]. The results from a study of *Bacillus* ACC-deaminase species (*B. methylotrophicus*, *B. siamensis*) in wheat showed the ability of PGPB to enhance wheat seed germination and seedling growth at different NaCl concentrations. This suggests the potential use of these species to improve crop growth in agricultural systems under salinity stress [59]. Furthermore, several studies confirmed the beneficial effects of halotolerant IAA-producing and halophilic bacterial strains on *Triticum aestivum* [60] and *Brassica napus* [61] under salinity stress conditions. These microorganisms were able to increase plant biomass, provide additional IAA uptake and induce salt tolerance by reducing the ETH levels.

Another recent study on *Brassica napus* reported the effect of seed inoculation with five different species of PGPB, i.e., *Azospirillum brasilense*, *Arthrobacter globiformis*, *Burkholderia fariambia*, *Herbaspirillum seropedicae* and *Pseudomonas* spp. (separately). Plants inoculated with PGPB showed increased development, reduced water loss due to low membrane damage, increased antioxidant activity and increased synthesis of osmolytic proline; moreover, no deleterious effects on their photosynthetic apparatus were reported [62]. Positive results were also obtained in a study on spinach (*Spinacia oleracea* L.). Inoculation of plants with halotolerant (*Pseudomonas* spp., *Thalassobacillus* spp., *Terribacillus* spp.) and chitinolytic (*Pseudomonas* spp., *Sanguibacter* spp., *Bacillus* spp.) strains improved plant growth and reduced salinity [63].

PGPB are also able to help plants to reduce salt stress through the formation of biofilms. Biofilms are one of the most important protective strategies against adverse and unpredictable environmental conditions. Sessile PGPB are therefore better able to survive and interact with plants than planktonic cells. In fact, these microorganisms were shown to be more resistant to antimicrobial compounds, drought and UV radiation [64]. Microorganisms can develop biofilms on a wide range of materials, such as roots and soil, improving crop and soil performance. In addition, increased EPS$_s$ production supports biofilm formation and improves tolerance to abiotic stresses such as salinity. EPSs-producing PGPB can bind Na$^+$ ions by reducing their availability, thereby reducing salt stress, increasing K$^+$ uptake and improving water uptake [65]. The production of EPSs also results in changes in the cell envelope, with an increase in water retention and the regulation of carbon sources [66].

This was confirmed by several studies conducted on different crops (barley, sunflower) [67]. In particular, sunflower plants inoculated with *Pseudomonas plecoglossicida* PB5 and *Bacillus licheniformis* AP6, two strains capable of forming biofilms, were more resistant to salt stress than non-inoculated plants [68].

The application of a silicon fertiliser (potassium silicate, K$_2$SiO$_3$) was shown to ameliorate the negative effects of various biotic and abiotic stresses on plants, including salinity [69]. Silicon can enhance salinity tolerance in plants by improving Na$^+$ and K$^+$ homeostasis, nutrient status, ROS-scavenging enzyme activity and photosynthetic efficiency [70–72]. Mahmood et al. [73] reported that treatment with PGPB combined with the foliar application of a silicon fertiliser led to better tolerance to salinity stress in mung bean plants compared to plants treated with PGPB or the silicon fertiliser alone. Subsequently, Al-Garni et al. [74] showed that the combined application of two strains of *Pseudomonas* (*Pseudomonas pseudoalcaligenes* and *P. putida*) with a silicon fertiliser alleviated the salinity stress in coriander by increasing relative water content, photosynthetic pigment concentrations,

peroxidase activity, total biomass, salt tolerance index and total phenolic content. The combined application of PGPB and a silicon fertiliser seems to be a feasible and promising strategy to improve plant performance in salt-affected farmlands [75]. The above strategies involving the use of PGPB and halotolerant bacteria are useful for remediating saline soils and improving plant growth under salinity stress.

*3.2. PGPB—Effects of Drought Stress*

Climate change is affecting crop production around the world. High temperatures combined with a lack of rainfall lead to drought, and this effect is more pronounced on marginal soils. Drought alters not only plant responses to pathogens, but also plant microbial communities. Drought stress, therefore, has a major impact on the quantity and quality of harvests, reducing the world's food supply, and is one of the most serious problems in agriculture [76]. It has been estimated that half of the world's arable land will be affected by drought in the first half of 2050, and this is closely linked to global temperature increases [77]. In several regions, and especially in semi-arid areas, the increase in frequency, duration and intensity of droughts, mainly driven by climate change dynamics, is expected to drastically reduce the current freshwater supplies, limiting crop development and yields, especially where agriculture is highly dependent on irrigation. The impact of climate change on water yields is already evident in drought-prone regions of the Mediterranean basin (Spain, Malta, Italy, Greece and Turkey). The hot and dry climate and the variability of rainfall intensity pose serious problems for the use of water resources in many regions of the Mediterranean basin, including Apulia. As far as the region of Apulia is concerned, the most vulnerable areas are located in the central and southern parts, which are characterised by a high percentage of vegetables and fruit trees (the most vulnerable crops). On the other hand, north-central Apulia (the Capitanata and Terre d'Apulia consortia) is less vulnerable, mainly due to the greater presence of vineyards and olive groves (more tolerant to water stress) [78].

Plants exposed to drought stress show different response mechanisms in the form of morphological, physiological and biochemical changes. In general, drought stress affects seed germination, plant growth and yield, transpiration rate, net photosynthesis rate, stomatal conductance, relative leaf water content and water potential [79–81].

Plants can use various strategies to avoid or tolerate water shortages, such as reducing transpiration and photosynthesis, enhancing the action of phytohormones and diverting energy into developing a stronger root system instead of producing new leaves [82].

Nitrogen, like many other micronutrients, is involved in several essential biochemical pathways that occur in plants, such as chlorophyll production and photosynthesis [83]. Plants have natural mechanisms to carry out N fixation, but this is a process that is highly susceptible to drought stress, leading to reduced growth rates [84]; furthermore, in the absence of water, nitrate reductase activity is reduced, resulting in poor uptake of the available nitrogen [85]. It is a common practice to use synthetic fertilisers to improve crop nitrogen uptake on drought-prone soils, but a sustainable alternative could be the use of PGPB. These bacteria regulate plant growth under drought stress conditions both directly (increased phytohormone production and nutrient availability) and indirectly (induction of systemic resistance (ISR), suppression of pathogens, synthesis of lytic enzymes) and secrete various compounds such as osmolytes, antioxidants, phytohormones, etc., that improve the osmotic potential of roots under drought stress conditions [86,87]. In drought-stressed and low-moisture soils, diazotrophic PGPB such as cyanobacteria, *Azospirilium* and *Azoarcus* [88] can produce and make nitrogen available to plants by enhancing nitrogen production, uptake and accumulation in plant tissues and soil. A recent study demonstrated the efficacy of a co-inoculum of *Bradyrhizobium japonicum* USDA110 and the PGPR *P. putida* NUU8 in soybean (*G. max* L.), showing increased nitrogen accumulation in plant tissues (+35%) and soil (+23%) compared to the control under drought stress conditions [89].

The critical response to biotic and abiotic stresses is based on the action of phytohormones, which slow down vital plant functions by reducing energy loss. Phytohormones

play a key role in regulating the plant response to biotic and abiotic stresses and include auxins, ETH, GA and ABA [90]. PGPB can modulate the levels of these hormones to improve drought stress resistance. One of the most important supporting activities of PGPB is the development of the root system to increase the uptake of water and macro- and micronutrients from the soil through the production of IAA [91]. Many studies demonstrated the production of these compounds by various microorganisms such as *Azospyrillium*, *Pseudomonas*, *Bacillus* and *Staphylococcus* spp. [92,93].

Wheat under drought conditions inoculated with *B. megaterium* and *B. licheniformis* showed different growth patterns due to the different production rates of IAA and ACC-deaminase, which were higher in *B. megaterium*. The study showed higher root system development in wheat inoculated with *B. megaterium* than in wheat inoculated with *B. licheniformis* [94].

Another study confirmed that *B. megaterium* applied to *Arabidospis taliana* showed a high rate of IAA production, and under drought stress conditions, plants inoculated with the microorganism showed a 1.2—3.0-fold increase in many plant growth parameters, such as fresh weight, dry weight, etc. PGPB activity of *B. megaterium* in maize, as demonstrated by Romero-Munar et al. [95], was enhanced when PGPB were co-inoculated with *Rhizophagus irregularis*, an arbuscular fungus, under drought and high-temperature stress conditions. The study showed that the double inoculation increased plant growth by 19%, while plants inoculated with *B. megaterium* alone showed an insignificant growth increase compared to the uninoculated control plants.

Another interesting compound produced by PGPB that may effectively assist IAA in modulating the response to phytohormones and, consequently, plant growth is ACC-deaminase. The rate of ACC-deaminase production by PGPB strains increases when stress conditions are more intense, as shown by Aguilera-Torres et al. [96]. PGPB isolated from two different sites at an altitude of 2050 m showed an interesting difference in ACC-deaminase activity, which was five-fold higher in the most stressed soil. The main activity of ACC-deaminase is the reduction of ethylene and related stress, and several studies demonstrated a positive relationship between ACC-deaminase activity and plant growth.

A study by Ojuederie et al. [97] on the growth-promoting activity of PGPB strains inoculated into maize under drought stress conditions emphasised the enhancing activity of ACC-deaminase. Among the three inoculated microorganisms, *Pseudomonas* sp. MRBP13 showed higher ACC-deaminase activity and stress reduction than the other microorganisms. In cluster bean (*Cyamopsis tetragonoloba* L.), increased ACC-deaminase activity was confirmed by ETH reduction. *Pseudomonas stutzeri* (AK17) and *Paenibacillus polymyxa* (KM6), which are producers of ACC-deaminase, led to lower ETH accumulation in inoculated plants than in non-inoculated controls. Furthermore, the activity of these PGPB increased the relative water content (RWC). RWC is a useful parameter for determining and monitoring the health of a plant under drought stress conditions and takes into account water uptake and water lost through transpiration [98].

An interesting characteristic of PGPB is the ability to produce EPSs. This trait, typical of some PGPB, can increase stress resistance under drought conditions by improving soil water retention [99]. In maize inoculated with *Bacillus velenenzis*, the combined effect of ACC-deaminase activity and EPS$_s$ production resulted in increased root length, fresh and dry weight and plant growth parameters [100].

The role of EPSs in alleviating drought stress was also investigated in two wheat cultivars, Johar-16 and Gold-16, using EPS$_s$-producing PGPB and IAA [101]. The strains tested were a *Chryseobacterium* sp. (LEW3), an *Acinetobacter* sp. (LEW9) and a *Klebsiella* sp. (LEW16). The plants inoculated with LEW16 had a larger root diameter than the uninoculated control for both varieties, and root growth was about 27% higher for both wheat varieties.

Further work and studies on the beneficial effects of PGPB on growth parameters and plant resistance under drought conditions are listed in Table 2.

**Table 2.** The effects of PGPB alleviating drought stress in plants. * (V: in vitro condition; F: field condition; Ghouse: greenhouse condition; Gchamber: growth chamber condition; planta: planta condition).

| PGPB | Plant Name | PGPB Mechanisms | Application Method * | Plant Response | Reference |
|---|---|---|---|---|---|
| *Providencia rettgeri* | *Hordeum vulgare* L. | ✓ Phosphate solubilisation<br>✓ EPS$_s$ and siderophore production | Seed inoculation (Ghouse) | ✓ Increased shoot dry weight<br>✓ 25% increase in relative water content | [102] |
| *Bacillus cereus* L90 | *Juglans regia* | ✓ CTK production<br>✓ Stimulation of antioxidant enzymes | Soil inoculation (Ghouse) | ✓ Increased base diameter and plant height<br>✓ Increased endogenous hormone production | [103] |
| *Bacillus velenenzis*<br>*Bacillus cereus*<br>*Pseudomonas baietica*<br>*Staphylococcus pasteuri* | *Triticum aestivum* L. | ✓ Production of IAA, EPSs, siderophore<br>✓ Phosphate solubilisation | Seed inoculation (Ghouse) | ✓ Shoots and roots development<br>✓ Increased in fresh and dry shoot and root weight | [93] |
| *Bacillus megaterium*<br>*Bacillus licheniformis* | *T. aestivum* L. | ✓ Production of EPSs, ACC-deaminase, siderophore production<br>✓ Antagonistic activity<br>✓ Potassium solubilisation<br>✓ Putative candidate proteins under drought stress | Seed inoculation (V, Ghouse) | ✓ Increased germination, shoot length, relative water content, antioxidant activity | [94] |
| *Pseudomonas aeruginosa*<br>*Enterobacter cloacae*<br>*Achromobacter xylosoxidans*<br>*Leclercia adecarboxylata* | *Z. mays* L. | ✓ ACC-deaminase production | Seed inoculation (V) | ✓ Increased shoot and root length<br>✓ Improved synthesis of chlorophyll a, chlorophyll b and total chlorophyll | [104] |

**Table 2.** *Cont.*

| PGPB | Plant Name | PGPB Mechanisms | Application Method * | Plant Response | Reference |
|---|---|---|---|---|---|
| *Pseudomonas azotoformans* | *T. aestivum* L. | ✓ EPSs, IAA, siderophore, HCN and NH$_3$ production<br>✓ Phosphate solubilisation | Seed inoculation (Gchamber) | ✓ Increased shoot and root length, number of roots, shoot and root fresh and dry weight, relative water content, root-adhering soil/root tissue ratio | [105] |
| *Pseudomonas* spp. *Serratia marcescens* | *T. aestivum* L. | ✓ EPSs, ACC-deaminase, siderophore and ammonia production | Seed inoculation (Ghouse) | ✓ Improved crop index<br>✓ Increased available micronutrients (Zn and Fe) | [106] |
| *Pseudomonas pseudoalcaligenes* | *Z. mays* L. | ✓ VOC production | Seed inoculation (V, Ghouse) | ✓ Enhanced production of photosynthetic pigments<br>✓ Increased production of phytohormones and antioxidant enzyme activity | [107] |

The studies presented in the table focused on the main crops typical of Apulia.

### 3.3. PGPB—Improvement of Plant Growth on Polluted Marginal Land

The 'health' of soils, particularly agricultural soils, is reflected in the health of consumers and is challenged every day by human activities that cause soil pollution. The deliberate or accidental introduction of hazardous substances into soil can alter the specific soil characteristics to such an extent that not only its protective functions but also its productive and ecological functions are impaired. Soil contamination may be localised in limited areas corresponding to known sources of contamination (contaminated sites) or may be due to inputs of contaminants whose origin cannot be identified or to the presence of multiple sources of pollution, e.g., agricultural practices, mainly related to the use of plant protection products, atmospheric emissions from industrial, civil and roadside installations and accumulation of nutrients in soils. The presence of contaminated sites is a problem common to all industrialised countries and results from the presence of anthropogenic activities that can lead to local soil contamination through spills, leaks from facilities/reservoirs and improper waste management. The latest State of the Environment study by the Regional Agency for Environmental Prevention and Protection (ARPA) showed that Apulia is still the region with the highest industrial emissions in Italy, despite the fact that air quality regulations (e.g., on dioxins) are stricter than in the rest of the country. Critical results were highlighted in the Taranto, Bari, Brindisi, Barletta–Andria–Trani and Foggia provinces. In Apulia, there are different sites of national importance where the environmental situation is serious, which include Manfredonia, Brindisi and Bari [108]. Significant levels of HMs, particularly chromium (Cr), were found in more than 400 hectares of soil in the Altamura and Gravina areas [109]. This contamination is the result of the inappropriate disposal of a wide range of wastes, the sources of which are likely to be of various origins: urban, industrial, hospital, agricultural, etc. The pollutants recognised by the Food and Agriculture Organization of the United Nations (FAO) are HMs, metalloids, radionuclides, synthetic organic compounds such as pesticides and polycyclic aromatic hydrocarbons, pathogenic bacteria and emerging pollutants such as pharmaceuticals and personal care products [110,111].

PGPB are involved in the soil remediation mechanism both indirectly, by promoting plant growth through phytoremediation [112–114], and directly, by activating internal cell mechanisms. Not all PGPB are able to biodegrade HMs, as shown in several studies that reported the negative effects of HMs on the physiology, metabolic activity and survival of microorganisms [115,116]. Resistance to metals is, of course, a necessary condition for the survival of microorganisms in contaminated soils and their restorative activity. The main mechanisms used by PGPB to restore heavy-metal-contaminated soils include detoxifying activities such as accumulation, precipitation, transformation. Bioaccumulation can be divided into two phases: bacterial binding to a heavy metal and subsequent HM inclusion into the bacterial cell through the cell membrane, resulting in a reduction in the contaminant levels in the soil [117]. Once inside the cell, through facilitated transport or passive diffusion, HMs are transformed or degraded to less toxic forms through oxidation, chelation or methylation [75].

Potential ameliorative abilities in the presence of heavy-metal stress by *Pseudomonas* spp., *Bacillus* spp., *Acinetobacter* spp., *Luteibacter* spp., *Azotobacter* spp., *Trichoderma* spp., etc., have been studied [118]. Arsenic stress-ameliorating activity of *Pseudomonas oleovorans* was observed in rice, as reported by Anand et al. [119]. The activity of the microorganism not only was effective for plant health but also was able to reduce the amount of As in soil and in rice shoots, roots and grains. A recent study demonstrated high arsenite (As(III)) and arsenate (As(V)) tolerance and efficient As(V) reduction and As(III) efflux activity in *P. putida* ARS1, an endophyte isolated from rice grown in arsenic-contaminated soil [120]. Genome sequencing revealed that *P. putida* ARS1 possesses two sets of chromosomal arsenic resistance genes (arsRCBH), which contribute to efficient As(V) reduction and As(III) efflux, resulting in high arsenic resistance. Furthermore, the co-culture of *P. putida*

ARS1 and *Wolffia globosa* (a strong arsenic accumulator with high potential for arsenic phytoremediation) increased arsenic accumulation in *W. globosa* by 69% and resulted in the removal of 91% of arsenic (from an initial concentration of 0.6 mg/L As(V)) from water within 3 days. As observed by Marwa et al. [121], arsenic contamination can also be bioremediated by other PGPB such as *Bacillus* spp. and *Acinetobacter* spp. *Bacillus flexus* and *Acinetobacter junii* are generally classified as PGPB due to their ability to solubilise phosphate and produce siderophores, IAA and ACC-deaminase; moreover, the strains showed growth in the presence of 150 mmol $L^{-1}$ As(V) and 70 mmol $L^{-1}$ As(III).

Enhancement activities were also evaluated in more critical situations, such as multi-metal contamination [122]. *Klebsiella* spp. M2 and *Kluyvera* spp. M8 showed detoxification activities in the presence of cadmium (Cd) and lead (Pb) by means of phosphate solubilisation mechanisms. The increase in available phosphate promoted the precipitation of Cd and Pb as phosphate compounds, reducing the heavy metal uptake by plant roots.

Mercury (Hg) is considered by the Agency for Toxic Substances and Disease Registry (ATSDR) to be the third most hazardous heavy metal found in contaminated soils [123]. More recently, a study by Chang et al. [124] focused on the mechanisms of Hg(II) resistance and sequestration by *Pseudomonas* sp. AN-B15. According to the authors, the main mechanisms involved are volatilisation and conversion of Hg(II) to mercury sulphide (HgS) and sulfhydryl mercury, but the biological pathways underlying these mechanisms are unknown.

With regard to direct bioremediation action against other HMs, the bibliography is limited, and the conducted studies focused on phytoextraction mechanisms and the support of PGPB in this bioremediation activity [125–128]. The bioremediation studies are not limited to HMs, but also concern the search for sustainable PGPB-mediated solutions against contamination by organic pollutants and pesticides. Particular attention has been paid to glyphosate, a widely used herbicide with known carcinogenic activities [129–131].

Very few studies have been conducted to assess the simultaneous growth-promoting and glyphosate-detoxifying effects of PGPB. A recent study compared eleven PGPB strains and selected five microorganisms capable of simultaneously enhancing maize (*Z. mays*) growth and degrading glyphosate at various concentrations [132]. *Enterobacter ludwigii*, *Pseudomonas aeruginosa*, *Klebsiella variicola*, *Enterobacter cloacae* and *Serratia liquefaciens* led to a reduction in glyphosate levels after 28 days under axenic conditions, using two concentrations of glyphosate, i.e., 100 and 200 mg/kg.

PGPB have also been tested for their ability to reduce the concentration of organic contaminants such as polycyclic aromatic hydrocarbons (PHAs) and gasoline by-products such as diesel in contaminated soils. In vitro and in vivo, *Bacillus marsiflavi* bac144 showed the ability to degrade hydrocarbons. In vitro, the microorganism was able to reduce the concentration of hydrocarbons by 65%, while in vivo, in maize, the degradation rate was increased by 46% [133]. Besides *Bacillus* spp., also *Pseudomonas* spp. degrade petroleum hydrocarbons. The detoxification effect investigated by Ambust et al. [134] is based on the ability of *Pseudomonas* spp. SA3 to produce a biosurfactant. As described by the authors, these microorganisms have the ability to reduce surface tension. This facilitates the reduction of hydrophobic compounds such as hydrocarbons.

## 4. Challenges in Marketing PGPB

As seen in the previous section, PGPB have enormous potential to be used as biofertilisers to replace chemical fertilisers/pesticides. The excessive and inappropriate use of synthetic fertilisers leads to pollution of the air and, especially, of groundwater through eutrophication; so, it is easy to see why it is important to give preference to biofertilisers. However, the global market for biofertilisers is only a small fraction of the market for synthetic pesticides due to their expensive production processes, inadequate storage stability, sensitivity to environmental factors, efficacy issues and other problems. Poor quality control, limited shelf life and lack of awareness are other barriers to the commercialisation

of biofertilisers [135]. The main problems related to the use and marketing of biofertilisers and possible solutions are listed below:

1.  *Short shelf life:* The limited shelf life of PGPB, which prevents biofertilisers from competing with synthetic pesticides, is one of the problems faced by farmers and manufacturers. This could be overcome by combining the inoculant with a carrier, which would not only prolong the viability of the inoculant, but also create ideal conditions for the rapid growth of the microorganisms when released in the field [136]. Other options include liquid biofertilisers, which have a longer shelf life (up to two years) compared to solid biofertilisers, which have a shelf life of six months. Another possible solution could be the encapsulation of PGPB in polymer matrices to protect the microorganisms introduced into the soil and ensure their slow and steady release.

2.  *Co-inoculation:* Scientific evidence suggests that it is preferable to co-inoculate different types of microorganisms, as together, they are more effective than individual PGPB strains when applied in the field [137]. There is evidence that some strains of bacteria can slow or block the development of other microorganisms, although it is true that within a consortium of microorganisms, one strain can support or complement the work of another [138]. In addition, different microbes may have different growth requirements, which can make growth synchronisation difficult. Therefore, a careful formulation of the biocontrol consortium is required to select the appropriate PGPB combination. In this particular scenario, endospores (such as *Bacillus* spp. and *Paenibacillus* spp.) are the right choice. Endospores confer greater stress resistance and stability on the formulation and storage of the inoculant and have a greater ability to survive under adverse conditions than other bacterial forms.

3.  *Competition in the rhizosphere and their impact on native bacterial communities:* The performance of PGPB in field trials is not always consistent with laboratory or greenhouse tests. This is often because PGPB are not competent in the rhizosphere. To be competent in the rhizosphere, they must be able to effectively colonise plant roots and to persist and multiply along plant roots for an extended period of time in the presence of the native microflora [139]. Furthermore, once inoculated onto a seed or a plant, PGPB, acting as invaders, can potentially induce changes in microbial communities, thus disrupting the niche previously established by the resident microbiota [140]. Three situations may occur: (i) PGPB can become stably established within the resident microflora and the community composition, (ii) soil resilience leads to the elimination of PGPB and the restoration of the baseline conditions, and (iii) PGPB can become established within the native microflora and then induce transient changes in the composition of the native microbial community, followed by the restoration of the baseline conditions [141,142]. If PGPB establish a stable interaction with the resident bacterial community, they may develop positive or negative relationships with the community members, leading to changes in the species composition of the community. These changes are not limited to the community level; cascading effects may extend to the ecosystem level, with unpredictable and possibly undesirable consequences for agroecosystem functioning [143]. For example, inoculation of plants with PGPB can lead to the so-called 'legacy effects' [144], which may include changes in the resident microbiome, nutrient cycling, disease suppression and organic matter persistence. It is important to note that the introduced PGPB may leave a functional legacy, whether or not they persist in the community. Further research is therefore needed to clarify how the introduction of PGPB may affect the structure and function of the resident community, of the ecosystem within the application area (e.g., cropland) and of adjacent ecosystems, as their effects are still largely unknown.

4.  *Public health:* Despite their enormous potential to promote plant development, pathogenic bacterial isolates can be harmful to humans. Of concern are *Pseudomonas* species, such as *P. fluorescens*, *P. putida*, *P. putrefaciens*, *P. stutzeri* and *P. pseudoalkaligenes*, and the opportunistic pathogen *P. aeruginosa* (which causes respiratory infections in

humans) [145]. Other *Bacillus*, *Ralstonia*, *Enterobacter*, *Acineto*, *Serratia*, *Rhodococcus*, *Klebsiella* and *Stenotrophomonas* species can be both plant growth promoters and human pathogens prevailing over beneficial and environmentally benign bacteria [145,146]. Assessing the pathogenicity of bacteria using biochemical and molecular tests can reduce the unintended use of pathogenic bacteria as biofertilisers.

5. *Field instability:* Success in the laboratory is often not accurately reflected in field trials. The main use of PGPB is their application in the field, although the development of a technology begins in the laboratory. Therefore, in order to evaluate and formulate a suitable PGPB inoculant, laboratory trials need to be associated with pot trials and then large-scale field trials. Few PGPB have been successfully registered for commercial use because of the inconsistency of the results between greenhouse and field trials. In addition, soil quality and climatic conditions are very important for the soil and plant colonisation of PGPB.

6. *Regulatory constraints:* Biofertilisers are subject to complicated product registration and patent application procedures. In addition, the regulatory processes tend to be very expensive, and the guidelines vary worldwide [147]. A globally coherent and coordinated regulatory policy is needed to standardise and facilitate the regulatory procedures for biofertilisers.

In conclusion, bio-fertilisers are a vital resource that, along with other sustainable agricultural practices, can help meet the challenge of feeding a growing global population at a time when agriculture faces various environmental pressures. It is true that biofertilisers may be considered more expensive and slower in producing effect than synthetic fertilisers, but by using biofertilisers it would be possible to grow healthy crops while improving sustainability and soil health.

## 5. Conclusions and Perspectives

Although the beneficial effects of PGPB have been demonstrated, further studies are needed to clarify exactly how effective they are depending on the type of culture and on soil and climatic conditions. Scientific evidence suggests that microorganisms isolated from the host plant microbiome are more effective than inoculations of non-endogenous microorganisms. The characterisation of the plant microbiome is therefore a key step in selecting the best strains. It is also important to bear in mind that no microorganism can be universally applied to every ecosystem or host plant; the choice of particular strains must be made taking into account the characteristics of the soil and the specific characteristics of the crop, in order to select the best microbial strain with the appropriate characteristics for each specific crop need. No less important is the competition for nutrients and ecological niches between the endogenous microbiota and the selected microbial strain, which could reduce the efficacy of the biostimulant.

Based on the research reviewed, it appears that PGPB are a sustainable and safe alternative to protect the environment and improve soil health by transforming degraded soils into healthy ones. High-quality crops can be produced in a sustainable and environmentally friendly way, helping to preserve the planet. Soils in the Mediterranean region are severely affected by degradation due to salinisation, high levels of desertification and poor land management. Unfortunately, as far as we know, there are still few studies on the use of PGPB in the Apulia region. However, several studies carried out in other areas on saline, arid and polluted soils, with crops typical of Apulia or well adapted to the Mediterranean climate of Apulia, have shown that their use is useful not only to improve yields, but also to increase the resilience of crops to abiotic, thermal and water stresses caused by climate change. Research has shown that microbial biostimulants based on PGPB formulations appear to be a better solution than the application of a single strain. In fact, these mixtures may be able to exert synergistic or additive biostimulatory effects. The choice of the microbial biostimulant product components is key to the effectiveness of a formulation. In order to make these formulations economically competitive on the commercial market and to achieve efficacy equal to or better than that of chemicals, their performance needs to be

improved by seeking solutions that can extend their efficacy and spectrum of action. In this respect, it may be useful to (i) select PGPB that can establish the greatest number of plant and rhizosphere relationships for use in the most diverse environmental conditions; (ii) select microorganisms with specific activities to achieve specific goals, such as increasing nutrient uptake when nutrients are scarce and re-establishing native plants in degraded soils using species-specific PGPB; (iii) use PGPB isolated from areas with salinity and desertification problems to increase plant tolerance to drought and salt stress; (iv) ensure the viability of microbial strains through the study of techniques that can guarantee their stabilization, such as immobilization; and (v) consider, from a purely economic perspective, issues regarding safety and economic viability, new perspectives and opportunities, changing legislation, public perception of the release of microbes into the wild, quality control requirements and patent eligibility. In conclusion, PGPB can be considered as plant probiotics to support the implementation of a sustainable intensification model of crop production, with the aims of increasing agricultural yields, conserving resources and reducing the negative impact of intensive agricultural practices on the environment, as well as reducing the impact of climate change on crop production. Most of the studies discussed in this review were conducted under greenhouse conditions and did not include field trials. However, field studies are needed, especially in the marginal areas of Apulia, to evaluate the effectiveness of PGPB in the promotion of plant development and tolerance to abiotic stresses.

**Author Contributions:** Conceptualization, A.R. and M.R.C.; investigation, A.R., A.d. and A.D.S.; writing—original draft preparation, A.R., A.d. and A.D.S.; writing—review and editing, A.R. and A.B.; funding acquisition, M.R.C., A.B. and M.S. All authors have read and agreed to the published version of the manuscript.

**Funding:** This study was carried out within the: Agritech National Research Center and received funding from the European Union Next-Generation EU (PIANO NAZIONALE DI RIPRESA E RE-SILIENZA (PNRR)—MISSIONE 4 COMPONENTE 2, INVESTIMENTO 1.4—D.D. 1032 17/06/2022, CN00000022). PON "Ricerca e Innovazione" 2014–2020 (PON R&I FSE-REACT EU) Azione IV.6 "Contratti di ricerca su tematiche Green". PROGETTO INTERDISCIPLINARE: INNOVAZIONE E DIGITALE Cluster "One Earth-One Health" (CUP D79J21011740006). This manuscript reflects only the authors' views and opinions; neither the European Union nor the European Commission can be considered responsible for them.

**Informed Consent Statement:** Not applicable.

**Data Availability Statement:** No new data were created for the production of this manuscript. All of the data here discussed and presented are available in the relative references here cited and listed.

**Conflicts of Interest:** The authors declare no conflict of interest.

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
