# Peer review of "Potential Use of Plant Growth-Promoting Bacteria to Enhance Growth and Soil Fertility in Marginal Areas: Focus on the Apulia Region, Italy"

_agronomy, doi:10.3390/agronomy13122983_

Round 1
Reviewer 1 Report
Comments and Suggestions for Authors
As a review article it focus on the important aspects of the use of PGPB under abiotic stress conditions, such as high salinty, drought and polution.
The manuscript titled: “Potential use of Plant Growth Promoting Bacteria to enhance growth and soil fertility in marginal areas: a focus on the Apulia region, Italy”
AS a review article it focuses on the important aspects of the use of PGPB under biotic and abiotic stress conditions. The article is well written and well organized. Authors should take into account a very few suggestions for changes found as follows:
Line 107 _ The authors should writte biofertilisers and not bio-fertilizes
Authors must also include in the text that biofertilizers are now designed by biostimulants /microbial biostimulants as european legislation has now designed.
Line 432 –Authors must write in full what ARPA means
Author Response
Response to Reviewer 1 Comments:
As a review article it focus on the important aspects of the use of PGPB under abiotic stress conditions, such as high salinty, drought and polution. The manuscript titled: “Potential use of Plant Growth Promoting Bacteria to enhance growth and soil fertility in marginal areas: a focus on the Apulia region, Italy”. As a review article it focuses on the important aspects of the use of PGPB under biotic and abiotic stress conditions. The article is well written and well organized. Authors should take into account a very few suggestions for changes found as follows:
Thank you very much for your comments and valuable suggestions. We are pleased that you liked our manuscript.
Line 107 _ The authors should writte biofertilisers and not bio-fertilizes
Modified
Authors must also include in the text that biofertilizers are now designed by biostimulants /microbial biostimulants as european legislation has now designed.
Thanks for the suggestion, but there is already a reference to the European regulation (line 78) in the text, and references to the Italian regulation (line 67) have also been included.
Line 432 –Authors must write in full what ARPA means
Added
Reviewer 2 Report
Comments and Suggestions for Authors
The article “Potential use of Plant Growth Promoting Bacteria to enhance growth and soil fertility in marginal areas: a focus on the Apulia region, Italy” authored by Racioppo is devoted to reviewing manuscripts on the role of PGPB in restoring the fertility of marginal lands. In recent years, the topic of PGPB in agriculture has been actively discussed, and a large number of experimental papers and reviews on this topic are published annually. A feature of this review, which distinguishes it from other reviews, is its concentration on a specific geographic region. Perhaps this difference could have been made clearer in the introduction.
There are several moments that, if corrected, could improve this article.
1. In the introduction, the authors first talk about the global problem of marginal land, then describe this problem in Italy, in particular in Apulia, and then again, without any transition, talk about the global problem. I would recommend that the authors restructure the text of the introduction, and first discuss the global situation with marginal lands and the use of PGPB, and then the situation in Italy.
2. The title suggests that you are focusing on the Apulia region, Italy, so in my opinion it would be worth emphasizing the importance of using PGPB specifically for this region in the “Conclusions and Perspectives” section. The work with PGPB that was carried out in Apulia can also be mentioned in Chapter 3.
3. Table 1. In recent years, a large number of articles have been published on the effects of PGPB on plants. Please explain on what basis you chose which articles to include in the table.
There are also a few minor comments that do not affect the overall quality of work
1. Please check the abbreviations. Abbreviations are introduced in several places in the text, and then full names are used.
2. L. 74-76: This sentence can be understood to mean that 98% of the world's food is produced in Apulia.
3. L.348. “the PGPR P. NUU8” – perhaps the species epithet is missing here
Author Response
Response to Reviewer 2 Comments:
The article “Potential use of Plant Growth Promoting Bacteria to enhance growth and soil fertility in marginal areas: a focus on the Apulia region, Italy” authored by Racioppo is devoted to reviewing manuscripts on the role of PGPB in restoring the fertility of marginal lands. In recent years, the topic of PGPB in agriculture has been actively discussed, and a large number of experimental papers and reviews on this topic are published annually. A feature of this review, which distinguishes it from other reviews, is its concentration on a specific geographic region. Perhaps this difference could have been made clearer in the introduction.
Thank you very much for your comments and valuable suggestions. As suggested, the introduction has been modified and the purpose of the review has been emphasised.
There are several moments that, if corrected, could improve this article.
- In the introduction, the authors first talk about the global problem of marginal land, then describe this problem in Italy, in particular in Apulia, and then again, without any transition, talk about the global problem. I would recommend that the authors restructure the text of the introduction, and first discuss the global situation with marginal lands and the use of PGPB, and then the situation in Italy.
Thank you for your valuable suggestion, we have restructured the text of the introduction and included first the part about the global situation of marginal lands and the use of PGPB and then the situation in Italy.
- The title suggests that you are focusing on the Apulia region, Italy, so in my opinion it would be worth emphasizing the importance of using PGPB specifically for this region in the “Conclusions and Perspectives” section. The work with PGPB that was carried out in Apulia can also be mentioned in Chapter 3.
Thank you for your suggestion. We have emphasised the importance of using PGPB specifically for the Apulia region in the "Conclusions and Prospects" section and in several parts of the text.
- Table 1. In recent years, a large number of articles have been published on the effects of PGPB on plants. Please explain on what basis you chose which articles to include in the table.
According to our knowledge and research, there are few works on the use of PGPB in Apulia. Therefore, we have included in Table 1 and 2 the most recent works carried out on crops typical of the Apulia region or on crops that would be well adapted to the climatic conditions of the region. This is explained in the text.
There are also a few minor comments that do not affect the overall quality of work
- Please check the abbreviations. Abbreviations are introduced in several places in the text, and then full names are used.
Modified
- 74-76: This sentence can be understood to mean that 98% of the world's food is produced in Apulia.
Modified
- 348. “the PGPR P. NUU8” – perhaps the species epithet is missing here
Added
Reviewer 3 Report
Comments and Suggestions for Authors
This article includes a review of scientific studies on the effects of Plant Growth-Promoting Bacteria (PGPB) on different crop plants. The authors aim to link this reality with another one, that of the loss of quality soil in southern Italy. Despite both topics being of interest, the final result is that there hasn't been a proper integration of both problems. The review primarily focuses on the effects of PGPB on plants, with few cases demonstrating that these bacteria improve the soil. I would suggest to the authors that they change the title of the work, reduce references to the worsening of soils in southern Italy, and instead concentrate on PGPB. The review is interesting, but the highlighted aspect distracts the reader's attention. In addition, the PGPB could aid both and plants from anywhere and it is no necessary to concentrate on Italy.
Although I recognize the value of the review related to PGRPs, I must add that it is somewhat simplistic, leaving out the effect of these bacteria on different root types such as proteoid or coralloid roots. This review should be expanded further to cover a wider range of plant types and ways of influencing other plant life.
Similarly, the review lacks information on the deleterious effects the same products from PGPB can have on different species or even on the same species at different concentrations. This deleterious effects as well as the competition between microorganisms for the roots of plants need to be presented as well in the review.
General comment: The fist time an acronym is used, it has to be given in full. The text contains many of those that need to be interpreted by the reader.

Author Response
Response to Reviewer 3 Comments:
This article includes a review of scientific studies on the effects of Plant Growth-Promoting Bacteria (PGPB) on different crop plants. The authors aim to link this reality with another one, that of the loss of quality soil in southern Italy. Despite both topics being of interest, the final result is that there hasn't been a proper integration of both problems. The review primarily focuses on the effects of PGPB on plants, with few cases demonstrating that these bacteria improve the soil. I would suggest to the authors that they change the title of the work, reduce references to the worsening of soils in southern Italy, and instead concentrate on PGPB. The review is interesting, but the highlighted aspect distracts the reader's attention. In addition, the PGPB could aid both and plants from anywhere and it is no necessary to concentrate on Italy.
This review highlighted the potential of PGPB and how it can be used to improve soil quality and agricultural production in marginal areas suffering from various problems related to excessive soil salinity, drought, soil pollution (...). It is known that PGPB, due to their functional properties (solution of minerals, for example) improve soil fertility. In fact, in this review were discussed several works that demonstrate how PGPB can improve the development of plants in salt soils or with low water. Unfortunately, as far as we know, work on the use of PGPB in the marginal areas of Apulia is few in the literature. It is an unexplored area, as there are no studies in the literature on the use of PGPB in these areas, with the aim of increasing cultivation or restoring the typical local crops lost in these areas, for example due to soil degradation. On the basis of this work, PGPB could therefore be a sustainable solution to increase production yields on soils affected by excessive salinity, drought or polluted soils, even in the marginal areas of Puglia affected by these problems. This is confirmed by a series of studies that show that the growth of plants inoculated with PGPB is higher than that of non-inoculate plants, both on soils affected by salinity, drought and contamination, and on soils not affected by these factors.
These aspects have been discussed in the objective of the work (lines 150 - 159).
Although I recognize the value of the review related to PGRPs, I must add that it is somewhat simplistic, leaving out the effect of these bacteria on different root types such as proteoid or coralloid roots. This review should be expanded further to cover a wider range of plant types and ways of influencing other plant life.
Thank you for your suggestion, but the review describes the mechanisms of action of PGPB in a general way to better understand their potential and ability to withstand stress, without distinguishing between the different types of roots, also because the focus of the review is different.
Similarly, the review lacks information on the deleterious effects the same products from PGPB can have on different species or even on the same species at different concentrations. This deleterious effects as well as the competition between microorganisms for the roots of plants need to be presented as well in the review.
Added in the “Challenges in marketing PGPB”
General comment: The first time an acronym is used, it has to be given in full. The text contains many of those that need to be interpreted by the reader.
Modified
In the attached file "agronomy-reviewer 3" find the responses to the comments given in the pdf file.

Round 2
Reviewer 2 Report
Comments and Suggestions for Authors
The authors revised the article and corrected all my comments. I believe that the article can be accepted for publication in its current form.
Author Response
Thank you for your comment.
Reviewer 3 Report
Comments and Suggestions for Authors
The authors have not introduced any modifications to the text. They justify maintaining the supposed specificity of the study by pointing out that there are no specific studies on the effect of growth-promoting bacteria in the Apulia region, which is equivalent to saying that there are none for the El Páramo region in the north west of Spain or for the wetbell of Australia, just to mention two areas. It is true that there are no specific studies in particular regions because an deeper analysis would have to be conducted for each soil. What helps advance science is finding generalities applicable to a broad spectrum of situations and then seeking solutions for specific cases. In the case of this manuscript, it would be very interesting to broaden the perspective to plant responses to microorganism treatments under Mediterranean climate conditions, which would add greater value to the work.
Author Response
We agree with this and it is perfectly in line with our manuscript. We are aware of the scarcity of work on PGPB carried out in Apulia region. Therefore, in the paper we have presented studies demonstrating plant responses to treatments with microorganisms under Mediterranean climatic conditions. In fact, all the studies selected and discussed have been carried out on crops that are typical of the Apulia region and of the Mediterranean area, or that are well adapted to the Mediterranean climatic conditions, as is also shown by some of the bibliographical references:
- Salicornia europaea : its distribution area is located from the Arctic, Mediterranean, and Subtropical regions (Constantin, C.G.; Zugravu, M.M.; Georgescu, M.; Constantin, M.F.; Mot, , A.; Paraschiv, M.; Dobrin, A. The Impact of the Growing Substrate on Morphological and Biochemical Features of Salicornia europaea L. Appl. Sci. 2023, 13, 10835. https://doi.org/10.3390/app131910835)
- Solanum lycopersicum : is among the vegetable crops mostly cultivated in the Mediterranean basin, especially in Italy , and Apulia region (Southern Italy) is particularly rich in agro biodiversity (Giuliani, M.M.; Gatta, G.; Nardella, E.; Tarantino, E. Water saving strategies assessment on processing tomato cultivated in Mediterranean region. Ital. J. Agron. 2016, 11, 69 76.; Cerasola, V.A.; Perlotti, L.; Pennisi, G.; Orsini, F.; Gianquinto, G. Potential Use of Superabsorbent Polymer on Drought Stressed Processing Tomato (Solanum lycopersicum L.) in a Mediterranean Climate. Horticulturae 2022, 8, 718. https://doi.org/10.3390/horticulturae8080718)
- Zea mays L. : Maize ( Zea mays L .) is a principal annual cereal crop that occurs as a maincomponent in the crop rotations in Mediterranean countries (Salama, H.S.A.; Nawar, A.I.; Khalil, H.E. Intercropping Pattern and N Fertilizer Schedule Affect the Performance of Additively Intercropped Maize and Forage Intercropped Maize and Forage Cowpea in the Mediterranean Region. Agronomy 2022, 12, 107. 2022, 12, 107. https://https://doi.org/10.3390/agronomy12010107 )
- Phaseolus vulgaris L. : common bean is typically cultivated in the Mediterranean basin (Alomari Mheidat, M.; Martín Palomo, M.J.; Castro Valdecantos, P.; Medina Zurita, N.; Moriana, A.; Corell, M. Effect of Water Stress on the Yield of Indeterminate Growth Green Bean Cultivars (Phaseolus vulgaris L .) during the Autumn Cycle in Southern Spain. Agriculture 2023, 13, 46. https://doi.org/10.3390/ agriculture13010046)
- Cotton (Gossypium hirsutum): Cotton can be cultivated in different regions of the Mediterranean (Can, F. et al. (2022) ‘Cotton growing around the Mediterranean.’, CABI Books. CABI. doi: 10.1079/9781800620216.0013). In Italy, Capitanata (northern part of Apulia ), is a suitable area for cotton cultivation because there is warmth and availability of water. The first organic cotton chain is born on the Gargano.
- Hordeum vulgare L.: is one of the more important cultivated crops in the Mediterranean regionis (María Guadalupe Arenas Corraliza, María Lourdes López Díaz, Víctor Rolo, Yonatan Cáceres, Gerardo Moreno, Phenological, morphological and physiological drivers of cereal grain yield in Mediterranean agroforestry systems, Agriculture, Ecosystems & Environment, Volume 340, 2022)
- Triticum aestivum L.: Wheat (Triticum aestivum L.) is the principle crop grown in many Mediterranean climate zones around the world (William F. Schillinger, Steven E. Schofstoll, J. Richard Alldredge, Available water and wheat grain yield relations in a Mediterranean climate, Field Crops Research, Volume 109, Issues 1 3, 2008, Pages 45 49, https://doi.org/10.1016/j.fcr.2008.06.008 )
